# Synthesis of New Liquid-Crystalline Compounds Based on Terminal Benzyloxy Group: Characterization, DFT and Mesomorphic Properties

**DOI:** 10.3390/molecules28093804

**Published:** 2023-04-28

**Authors:** Alaa Z. Omar, Mohammed L. Alazmi, Mai S. Alsubaie, Ezzat A. Hamed, Hoda A. Ahmed, Mohamed A. El-Atawy

**Affiliations:** 1Chemistry Department, Faculty of Science, Alexandria University, P.O. Box 426 Ibrahemia, Alexandria 21321, Egypt; alaazaki@alexu.edu.eg (A.Z.O.); mohamed.elatawi@alexu.edu.eg (M.A.E.-A.); 2Department of Chemistry, Faculty of Science, Cairo University, P.O. Box 12613, Giza 12613, Egypt; 3Chemistry Department, Faculty of Science, Taibah University, Yanbu 46423, Saudi Arabia

**Keywords:** benzyloxy liquid crystals, azomethine, smectic A, DFT, mesophase

## Abstract

The effect of the terminal benzyloxy group on the mesomorphic properties of liquid crystalline materials developed from rod-like Schiff base has been described. For this objective, a novel Schiff base liquid crystal family, specifically new series of Schiff base liquid crystals, namely, (*E*)-4-(alkyloxy)-*N*-(4-(benzyloxy)benzylidene)aniline, **I_n_**, are prepared and investigated in detail. The length of the terminal alkyloxy chain (*n*) varies amongst the compounds in the series. Where *n* varies between 6, 8 and 16 carbons. At the other end of the compounds, benzyloxy moiety was attached. The molecular structures of all synthesized compounds were established using different spectroscopic techniques. The molecular self-assembly was explored using differential scanning calorimetry (DSC) and polarized optical microscope (POM). Depending on the length of the terminal alkyloxy chain, only one type of SmA phase with different stability was observed. The previously reported para-substituted systems and the present investigated compounds were compared and discussed. The calculated quantum chemical parameters were computationally correlated using the DFT method via the B3LYP 6-311G(d,p) basis set. The theoretical computations revealed that the length of the alkyl side chain influences the zero-point energy, reactivity and other estimated thermodynamic parameters of benzoyloxy/azomethine derivatives. Furthermore, the FMO energy analysis shows that molecule **I_16_** have higher HOMO energies than the other compounds, and **I_6_** has a much lower LUMO level than the rest.

## 1. Introduction

Tools for structure-activity relationships have become necessary to design new materials in order to attain the right properties for device applications [1,2,3,4,5,6]. Applications for liquid crystal (LC) instrumentation include optical devices and temperature/humidity sensors [7,8,9,10,11]. Therefore, anisotropic mesogenic form variables and principles are used to develop liquid-crystal molecular structures. There have been several two- and three-ring compounds based on Schiff base/ester liquid crystals have been examined, and their optical properties were investigated in order to better understand the relationship between the mesogens’ molecular geometry and their mesomorphic characteristics [12,13,14,15,16]. In LCs, the molecules are organized in a variety of ways that result in phases with varying degrees of order and symmetry between the crystalline and isotropic phases [17]. Since LCs materials have many valuable uses in industrial, scientific and medical technologies, this specific phase of matter differs from the three most well-known matter phases (solid, liquid and gas). Particularly important technologically for display devices and other uses are thermotropic LCs [18,19]. LC phases are regarded as the most promising prospects for the development of innovative high-performance components that are frequently used in a wide range of demanding applications, including thermoplastics [20], high-performance fibres [21], optical data storage applications [22] and light composite in the aerospace industry [23]. The most well-known examples are the rod-like calamitic LCs (polar or non-polar) generated by joining flexible alkyl tails with rigid cores such as aliphatic, aromatic, or heteroaromatic rings through linking groups. These molecules were fluidly self-assembled into various layered smectic phases or non-layered nematic phases. Due to their extensive use in optical and electrooptical devices, LCs have earned a distinct place in materials research due to their peculiar mesogenic property [24,25,26]. Due to their technological significance and the need to learn the fundamentals of LC research, a variety of polar LC compounds bearing a terminal polar group, including CN, NCS, NO_2_, F, CF_3_ and Br, have been created, and their physical properties have been assessed [27,28]. In some recent investigations, we have systematically created new LCs by altering the linking group and number of aromatic rings in the core part with one or more polar terminal groups, and we have characterized their mesogenic properties as well as the structural changes of the molecule at phase transitions [29,30,31].

The hard core of numerous low molar mass calamitic mesogenic compounds usually consists of many 1,4-disubstituted phenyl rings, some of which may or may not have lateral substituents, the terminals on one or both sides are linked to a long flexible hydrocarbon chain (with or without halo substituents). By making structural changes to LCs and/or adding fluorine or other small-size lateral substituents to the rigid core, one can change how mesomorphic they are [32]. Thermotropic LCs are mesogenic compounds whose mesomorphism is solely dependent on changes in the ambient temperature. In nematic thermotropic LCs, the longitudinal molecular axes align themselves parallel to the director axis, exhibiting a long-range orientational order. The outlined orientational order, as well as a translational order in which the molecules are ordered on equidistant planes to generate a periodic layered structure with a particular layer thickness, coexist in smectic thermotropic LCs [33].

The selection of a mesogenic core, terminal groups and a flexible chain are the essential specification in the design of new thermotropic LCs [16,29,30,31,34,35,36,37,38,39]. A para-substituted phenyl ring is used to confirm that the molecules have structural linearity and large molecular polarizability [40]. The mesophase behaviour of compounds of type (*E*)-3(or4) -(alkyloxy)-*N*-(trifluoromethyl)benzylideneaniline was previously investigated in our lab using a terminal polar trifluoromethyl substituent [40] as well as another terminal alkoxy side group of varying chain length. In that study, [40] found that the polarity and/or polarisability of the mesogenic portion of the molecule, which is indisputably affected by the polarity of the substituent, would consequently affect the polarity of the entire molecular structure by the way the polar substituents interact mesomerically with the rest of the molecule, enhanced the stability of the mesophase. A later study expanded the [41] investigation to include more compounds of type (*E*)-4-(alkoxy)-*N*-(4-(methylthio)benzylidene)aniline [41], where the thioether moiety was connected to the terminal ring. All thioether derivatives with liquid crystal phases, with the exception of the compound with the longest chain, exhibits non-mesomorphic behaviour, according to the study [41].

In the present study, we have synthesized a new polar LC system, (*E*)-4-(alkyloxy)-*N*-(4-(benzyloxy)benzylidene)aniline, in which the azomethine linking group and two terminal groups of ether benzyloxy and long side chain alkoxy. The azomethine linking group incorporation has been performed for many other LC systems [29,30,31], providing more conformational flexibility, which affects the mesomorphic behaviour. The different phases are identified by the DSC and POM. All possible structures are optimized using DFT to correlate with experimental results.

## 2. Results and Discussion

### 2.1. Chemistry

Schiff base linker groups provide several advantages when used as a connecting group in liquid crystal molecules. They can be easily synthesized and incorporated into a wide range of mesogenic structures. In addition, the imine bond is relatively stable, ensuring the stability of the liquid crystal phase over a wide range of temperatures [42]. Moreover, the linker group can be tailored to modify the liquid crystal properties, such as the phase transition temperature and the electro-optical response. Finally, one of the key properties of azomethine linker groups is their ability to undergo light-induced isomerization, altering their shape and orientation within a liquid crystal phase. As shown in Figure 1, Schiff bases **I_n_** were prepared by directly condensing 4-benzyloxybenzaldehyde with alkyloxy anilines [43]. Both Fourier transform infrared (FT-IR) and nuclear magnetic resonance (NMR) spectroscopy demonstrated that all the produced compounds possessed the aliphatic proton of the benzyloxy and terminal alkyloxy chain. The compounds’ molecular formulas and molecular structures were validated using elemental analysis, FT-IR measurements and NMR spectroscopy. The spectroscopic findings supported the predicted structures. Compounds **I_n_** infrared spectra showed absorption bands in the range from 2918 to 2931 cm^−1^, which are consistent with aliphatic C-H stretching’s absorption. The imine linker group also had an absorption band of about 1615 cm^−1^, which is the typical stretching of the C=N.

The terminal alkyloxy side chain protons appeared in the H-NMR spectrums with chemical shifts ranging from 0.81 to 3.98 ppm, whereas the protons of the methylene group, which corresponds to the benzyloxy group, appeared from 5.06 to 5.20 ppm. Furthermore, their corresponding saturated carbons revealed carbon NMR signals in the range of (14.6–69.8 ppm). Additionally, the aromatic rings have been identified in both proton and carbon NMR spectra. As a result, aromatic protons displayed multiplet signals in ^1^H-NMR at downfield frequencies ranging from 6.95 to 7.87 ppm. In addition, the ^13^C-NMR spectra revealed downfield signals corresponding to aromatic ring carbons in the range from 114 to 157 ppm. The quaternary aromatic carbon that bears the benzyloxy and alkyloxy side chain has been ascribed to the highest deshielded peaks near 161–158 ppm. Moreover, NMR was used to establish the existence of the azomethine group. As a result, singlet signals in proton NMR spectra were seen around 8.33–8.55 ppm, which corresponds to the proton of azomethine. Moreover, the carbon NMR displayed a downfield signal of about 144–145 ppm, which was attributed to the azomethine groups’ unsaturated carbon. Finally, mass spectrometry confirmed the corresponding molecular weight for all compounds under investigation.

### 2.2. Optical and Mesomorphic Properties

All of the synthetic materials’ transition temperatures and associated enthalpies from DSC measurements are listed in Table 1 and graphically depicted in Figure 1. The repeatability of the heating and cooling DSC curves proved that all derivatives were thermally stable. Figure 2a–c illustrates the DSC heating and cooling traces of all compounds **I_n_**.

As shown in Table 1, all produced compounds have mesomorphic phases. Additionally, at different temperatures, each of them exclusively exhibits the same type of mesophase with the LC phase, with the thermal stability varied in accordance with the length of the terminal alkoxy substituent. Furthermore, a characteristic image for a SmA phase displayed by conventional rod-like LCs was seen under crossed polarizers. 

Except for the short-length molecule **I_6_**, all derivatives from the **I_n_** series exhibit enantiotropic LC phases, as seen in Table 1. Moreover, each of them only displays the same type of mesophase with LC phase at various temperatures, with the thermal stability varying, according to the length of the terminal alkoxy substituent. For example, on cooling the first member of this series **I_6_** (with *n* = OC6H_13_) under crossed polarizers, a typical texture was observed for a smectic A (SmA) phase exhibited by conventional rod-like LCs (see Figure 3a). On further cooling of **I_6_**, the SmA phase remains lower ~ 107.0 °C, which is a relatively wide range till a direct transition to the crystal state occurs at *T* ~ 108.0 °C without the formation of any additional types of LC phases till crystallization. The recorded enthalpy value for the Iso-SmA transition is Δ*H* ~ 5.5 kJ/mol.

For the next derivative, **I_8_** having X = OC_8_H_17,_ the melting temperature is reduced, and the SmA phase is observed enantiotropically at the higher temperature LC phase. This stability of the formed phase is characterized by higher viscosity, meaning a higher degree of order (SmA phase). This was also supported by the higher value of transition enthalpy recorded for the SmA-Iso transition (Δ*H* ~ 5.7 kJ/mol) that was recorded at *T* ~ 129.0 °C.

The third compound of the group, **I_16_**, is created by changing the terminal alkoxy chain from an octyloxy chain to a hexadecyloxy chain. Figure 1 shows that due to the diluting effect brought on by the longer alkoxy chain, the melting temperatures are lowered to 118.5 °C. In general, **I_16_** exhibits the same phase type. Surprisingly, this slight modification widens the range of the SmA phase. Therefore, the SmA phase of **I_16_** is observed with a mesomorphic range and stability of 17 and 135.5 °C, respectively.

### 2.3. Comparison with Related Materials

To understand the effect of the terminal benzyloxy substitution on the phase behaviour of the investigated compounds, it is interesting to compare them with related materials reported before with different substitutions having the same aromatic core of compounds **I_n_** but without trifluoromethyl substitution (**II_n_**) [40] and thioether moiety (**III_n_**) [41]. The number of carbon atoms in the terminal chain (*n*) in all compared series also 6, 8 and 12 as those used in compounds **I_n_** (Figure 4).

As can be seen from the last reported data [41], all compounds are mesomorphic, having either monotropic or enantiotropic LC phases depending on the nature of the terminal substituent, except the III_n_ derivative shows a non-mesomorphic behaviour. Both homologues II_n_ and III_n_ have the same type of mesophase, as only the N phase is observed for both series (except III_n_). However, in the case of the benzyloxy compounds I_n_ the melting point is increased, resulting in a wider range of more ordered SmA phases covering all side chain lengths. It can be concluded that the strong dependence of the LC phase-type on the terminal substitution. Therefore, for trifluoromethyl and thioether compounds, only N phases were observed and on replacing the terminal side by the benzyloxy group, the mesomorphic properties are totally removed from the N phases, and the SmA phases only are formed. 

Overall, this comparison indicates that benzyloxy derivatives result in more ordered mesophases due to their rigidity and, therefore, the SmA phases are replaced with N phases in addition to the stabilization of the SmA phases in most cases.

### 2.4. DFT Study

#### 2.4.1. Optimized Geometry and Thermal Parameters

Density functional theory (DFT) is a popular theoretical method used to study the molecular geometry of liquid crystals. In this approach, the electronic structure of the molecule is calculated based on the principles of quantum mechanics. The DFT method considers the electron density distribution of the molecule, as well as the interactions between the atoms and molecules. By analysing the electronic structure of the liquid crystal, DFT can provide information about the molecular arrangement of the liquid crystal. This includes determining whether the molecules are planar, meaning all atoms lie on a single plane or non-planar. Experimental techniques, such as X-ray scattering, can also be used to determine the molecular planarity in liquid crystals. However, DFT calculations offer a more detailed understanding of the molecular structure and can help to design and optimize new liquid crystal materials with desirable properties.

The molecular planarity of liquid crystals is a key factor in determining their physical properties as melting point, viscosity and electrical conductivity. This is because planarity influences molecular packing, intermolecular interactions and the strength of the intra- and inter-molecular forces that hold the liquid crystal together.

In addition, the planarity of liquid crystals is important in various applications such as LCDs, electro-optic devices and medical sensors. Planar liquid crystal tends to align in a specific direction, creating an ordered structure, which allows them to have anisotropic (direction-dependent) optical properties, such as polarized light transmission, which is used in various display technologies. Additionally, the planar alignment of liquid crystals is crucial for the functioning of liquid crystal display (LCD) technology, which is widely used in modern electronics such as televisions, laptops and smartphones. Moreover, the planarity allows the molecules to respond to an electric field, thereby altering their orientation and changing the way light is transmitted through the material. Optimized molecular structures of the investigated compounds **I_n_** using the B3LYP 6-311G(d,p) basis set [44,45] are presented in Figure 5. The thermodynamic parameters, as well as the energetics, are listed in Table 2. As illustrated in Figure 5, all molecular configurations are planar; accordingly, the central aromatic ring forms a dihedral angle of ≈ 2.8° and 3.0° with respect to the terminal benzyloxy group and the other terminal aryl ring, respectively. This shows that there is almost no deviation from planarity over the entire molecule. Furthermore, the outcomes also showed that altering the length of the terminal alkyloxy group had no appreciable impact on molecular geometry and planarity. As a result, the molecular planarity of the examined compounds **1n** enhances and improves molecular packing in the condensed liquid crystalline phase. It should be mentioned that although our findings provide an estimation of the molecular structure in the gas phase, the presence of these compounds in the condensed liquid crystalline phases may exhibit some differences.

It was revealed that the length of the alkyl chain influences the zero-point energy and other computed thermodynamic characteristics of benzoyloxy/azomethine derivatives [30,31]. They were projected to increase as the length of the alkyl chain in series **I_n_** increased. The extended nonpolar moiety gives great thermal stability, as observed in compounds **I_16_**, whereas the compound with a smaller alkyl chain **I_6_** has the lowest thermal stability, Table 2.

Both dipole moment and polarizability play crucial roles in determining the properties of liquid crystalline materials. The molecular interactions generated by these properties result in the unique structure and behaviour of liquid crystal phases, making them important for a range of applications in electronics, displays and optical devices.

The dipole moment is a measure of the separation of positive and negative charges in a molecule. Molecules with a large dipole moment tend to interact strongly with each other, resulting in higher intermolecular forces and increased ordering. Liquid crystalline materials with high dipole moments display strong dipolar interactions, resulting in increased order and stability.

On the other hand, polarizability is a measure of how easily a molecule can be distorted by an external electric field. Polarizable molecules exhibit strong intermolecular interactions, resulting in a high degree of ordering in liquid crystalline materials. The polarizability of a molecule is influenced by its size, shape and electronic distribution, with larger molecules and those with a more symmetrical shape exhibiting higher polarizability.

The DFT-calculated dipole moments and molecule polarizability for the homologue series **I_n_** computed at the same level of theory are presented in Table 2. The outcomes revealed that as the terminal alkoxy chain length increased, the polarizability did as well, ranging from 348.111 to 452.945 Bohr3. As the polarizability rises, the dispersion forces become stronger, boiling and melting temperatures rise accordingly. Moreover, the DFT calculated dipole decreases as the chain length of the terminal nonpolar alkyl group increases. Thus, the length of the terminal alkyl chain has a substantial impact on both the dipole moment and polarizability of the studied compounds **I_n_**.

#### 2.4.2. Frontier Molecular Orbitals

The energy of the frontier molecular orbitals of liquid crystal materials, namely, highest-occupied molecular orbital (HOMO) and lowest unoccupied molecular orbital (LUMO), plays an important role in determining their optical, electrical and mechanical properties. By optimizing the energy levels of the frontier molecular orbitals, it is possible to tailor the properties of liquid crystal materials for specific applications such as displays, sensors and optical communication devices.

The HOMO energy level of a liquid crystal molecule determines its electron-donating ability. Higher HOMO energy levels indicate stronger electron-donating ability, which can affect the conductivity of liquid crystal materials. In addition, the HOMO energy level can also influence the thermal stability and the ability of the material to align in an applied electric field. On the other hand, the LUMO energy level of a liquid crystal molecule determines its electron-accepting ability. Lower LUMO energy levels indicate stronger electron-accepting ability, which can affect electron mobility and the response time of liquid crystal materials. Additionally, the LUMO energy level can also influence photoconductivity and the ability of the material to undergo photorefractive changes.

As shown in Table 3. According to the DFT calculations using the B3LYP 6-311G(d,p) basis set, in the case of the homologues series **I_n_**, the length of the terminal alkoxy chain has no impact on the energy values of the FMO. The energy gap (ΔE ≈ 3.96) is thus essentially the same for all of the molecules examined and is unaffected by the length of the terminal alkoxy chain. This relatively small energy gap for all the investigated compounds displays that these compounds are soft and reactive [46,47]. Additionally, as presented in Figure 6, neither the terminal benzyl group nor the terminal alkyl chain participates in either the HOMO or LUMO electron densities. The HOMO and LUMO are mainly localized over the central ring, imine linker and the terminal ring bearing the alkyloxy group.

#### 2.4.3. Chemical Reactivity Descriptors

Furthermore, absolute electronegativity, χ, chemical potentials, μ, absolute hardness, η, absolute softness, σ, global electrophilicity, ω, global softness, S and additional electronic charge, ΔN_max_, have been calculated from the HOMO and LUMO energies (Table 3) to give indications about the reactivity and stability of the molecules studied [48,49].

Table 3 points out that, Because **I_6_** has the greatest absolute electronegativity, it is more likely to form stable liquid crystal phases and to have stronger intermolecular interactions. Additionally, **I_6_** has the greatest global electrophilicity, making it more reactive toward other molecules and perhaps involved in chemical reactions that change the liquid crystal’s characteristics. In liquid crystals, the chemical potential is an important parameter because it determines the equilibrium between the liquid crystal and other phases, such as the isotropic phase. A molecule with a high chemical potential will be more likely to remain in the liquid crystal phase, while a molecule with a low chemical potential will be more likely to transition to the isotropic phase. ΔNmax measures the number of electrons that can be transferred from the molecule in reaction, soft molecules would have a large value of ΔNmax.

#### 2.4.4. Molecular Electrostatic Potential (MEP)

The molecular electrostatic potential of liquid crystalline materials plays a vital role in determining their properties, ranging from their stability, optical properties, to electronic conductivity. Understanding the impact of molecular electrostatic potential can help guide the design of new liquid crystalline materials with specific properties.

In liquid crystalline materials, the orientation and alignment of the molecules are critical in determining their properties. The electrostatic interaction between the molecules is directly related to the stability of the material. The higher the intermolecular electrostatic interaction is, the more stable the liquid crystal will be.

Moreover, the molecular electrostatic potential also impacts the optical properties of the material. The alignment of the molecules can lead to anisotropic behaviour of the material, influencing the refractive index and birefringence. The molecular electrostatic potential contributes to this behaviour by orienting the molecules in specific patterns and influencing their interaction with light.

Furthermore, the electronic conductivity of liquid crystalline materials is also influenced by their molecular electrostatic potential. In certain types of liquid crystal materials, such as those containing conductive organic molecules, the orientation and alignment of the molecules significantly influence the electron transfer and conductivity properties of the material.

Figure 7 shows the MEP of the benzoyloxy/azomethine derivatives **I_n_**, which reveals that the shadowing of polar atoms comprising oxygen and nitrogen atoms by a red cloud indicates a high electron density for these locations. The green cloud, on the other hand, is primarily spread over the alkyl chains and suggests low electron density but strong electrostatic potential.

## 3. Conclusions

Herein, we synthesize and thoroughly study a new family of Schiff base liquid crystals called (*E*)-4-(alkyloxy)-*N*-(4-(benzyloxy)benzylidene)aniline. Compounds have a benzyloxy moiety linked to one extreme side of the molecule. Using different spectroscopic methods, the molecular structures of all synthesized compounds were confirmed. Using DSC and POM, the molecule self-assembly was investigated for all designed derivatives. The B3LYP 6-311G(d,p) basis set of the DFT method was used to computationally correlate the estimated quantum chemical parameters. The study revealed the following:Except for the shortest chain derivative **I_6_**, all synthesized compounds show enantiotropic temperature mesomorphic ranges;The presently investigated compounds and the previously reported trifluoromethyl and thioether systems were compared and revealed to the benzyloxy group impact the smectogenic properties are totally removed from the N phases;Both the length of the alkoxy chain and the rigidity of the terminal benzyloxy substituent have a significant impact on the geometrical characteristics of the formed compounds;By enhancing the lateral attraction, smectic mesophase is improved as the dipole moment increases;The electrical properties of the terminal substituent had an effect on the FMOs energy gap and the global softness (S).

## 4. Experimental

### 4.1. Instruments and Apparatus

For more details, see Appendix A.

### 4.2. General Method for Synthesis of (E)-4-(alkyloxy)-N-(4-(benzyloxy)benzylidene)aniline **I_n_**

Alkyloxy anilines (0.01 mol) were added to a stirred solution of 4-(benzyloxy)benzaldehyde (0.01 mol) in 10 mL absolute ethanol, and the mixture was allowed to reflux for two hours. The TLC was used to monitor the reaction mixture. The crude products were filtered and washed with absolute ethanol.

(*E*)-4-(hexyloxy)-*N*-(4-(benzyloxy)benzylidene)aniline **I_6_**

Colourless crystals, (93%) yield; m.p. 135 °C. IR (KBr): ῡ 3037 (S_P_^2^ =C-H), 2931 (S_P_^3^ -C-H) and 1614 (C=N) cm^−1^.^1^H NMR (DMSO-*d6*, 400 MHz): δ 8.54 (s, 1H, CH=N), 7.86 (d, *J* = 8.4 Hz, 2H, Ar-H), 7.48 (d, *J* = 7.4 Hz, 2H, Ar-H), 7.42 (t, *J* = 7.4 Hz, 2H, Ar-H), 7.38–7.32 (m, 1H, Ar-H), 7.24 (d, *J* = 8.6 Hz, 2H, Ar-H), 7.14 (d, *J* = 8.4 Hz, 2H, Ar-H), 6.95 (d, *J* = 8.6 Hz, 2H, Ar-H), 5.20 (s, 2H, OCH_2_Ph), 3.97 (t, *J* = 6.4 Hz, 2H, OCH_2_), 1.70 (m, 2H, CH_2_), 1.43 (m, 2H, CH_2_), 1.32 (m, 4H, 2 CH_2_) and 0.89 (t, *J* = 6.4 Hz, 3H, CH_3_) ppm. ^13^C NMR (DMSO-*d6*, 101 MHz): δ 161.14, 158.08, 144.80, 137.16, 130.60, 129.86, 128.97, 128.45, 128.28, 127.86, 122.69, 115.54, 115.39, 69.87, 68.12, 31.48, 29.15, 25.67, 22.55 and 14.43 ppm. C_26_H_29_NO_2_ requires: C, 80.57; H, 7.55; N, 3.61% found: C, 80.41; H, 7.72; N, 3.87%. MS M^+^ at *m*/*z* 387.44 (34%).

(*E*)-4-(octyloxy)-*N*-(4-(benzyloxy)benzylidene)aniline **I_8_**

Colourless crystals, (90%) yield; m.p. 128 °C. IR (KBr): ῡ 3062 (S_P_^2^ =C-H), 2925 (S_P_^3^ -C-H) and 1615 (C=N) cm^−1^. ^1^H NMR (DMSO-*d6*, 400 MHz): δ 8.55 (s, 1H, CH=N), 7.87 (d, *J* = 8.2 Hz, 2H, Ar-H), 7.49 (d, *J* = 6.9 Hz, 2H, Ar-H), 7.43 (t, *J* = 7.3 Hz, 2H, Ar-H), 7.37 (t, *J* = 7.3 Hz, 1H, Ar-H), 7.24 (d, *J* = 6.9 Hz, 2H, Ar-H), 7.15 (d, *J* = 6.8 Hz, 2H, Ar-H), 6.96 (d, *J* = 8.9 Hz, 2H, Ar-H), 5.20 (s, 2H, OCH_2_Ph), 3.98 (t, *J* = 6.6 Hz, 2H, OCH_2_), 1.81–1.65 (m, 2H, CH_2_), 1.48–1.37 (m, 2H, CH_2_), 1.38–1.11 (m, 8H, CH_2_) and 0.88 (t, *J* = 6.6 Hz, 3H, CH_3_) ppm.^13^C NMR (DMSO-*d6*, 101 MHz): δ 158.07, 157.80, 144.47, 142.78, 133.33, 129.25, 128.79, 128.54, 127.72, 125.86, 122.84, 115.54, 115.44, 69.85, 68.13, 31.72, 29.22, 29.18, 29.15, 26.01, 22.56, 14.64 and 14.45 ppm. C_28_H_33_NO_2_ requires: C, 80.91; H, 8.01; N, 3.37% found: C, 80.67; H, 8.23; N, 3.59%. MS M^+^ at *m*/*z* 415.13 (22%).

(*E*)-4-(hexdecyloxy)-*N*-(4-(benzyloxy)benzylidene)aniline **I_16_**

Colourless crystals, (88%) yield; m.p. 119 °C. IR (KBr): ῡ 3037 (S_P_^2^ =C-H), 2918 (S_P_^3^ -C-H) and 1615 (C=N) cm^−1^. ^1^H NMR (CDCl_3_, 400 MHz): δ 8.33 (s, 1H, CH=N), 7.76 (d, *J* = 8.4 Hz, 2H, Ar-H), 7.38 (d, *J* = 6.2 Hz, 2H, Ar-H), 7.33 (t, *J* = 6.6 Hz, 2H, Ar-H), 7.27 (t, *J* = 7.1 Hz, 1H, Ar-H), 7.12 (d, *J* = 8.5 Hz, 2H, Ar-H), 6.98 (d, *J* = 7.1 Hz, 2H, Ar-H), 6.84 (d, *J* = 8.5 Hz, 2H, Ar-H), 5.06 (s, 2H, OCH_2_Ph), 3.89 (t, *J* = 6.3 Hz, 2H, OCH_2_), 1.79–1.65 (m, 2H, CH_2_), 1.52 (s, 4H, 2CH_2_), 1.45–1.33 (m, 4H, 2CH_2_), 1.29–1.12 (m, 18H, 9CH_2_) and 0.81 (t, *J* = 6.3 Hz, 3H, CH_3_) ppm. ^13^C NMR (CDCl_3_, 101 MHz): δ 161.22, 157.68, 157.60, 145.04, 136.56, 130.23, 129.77, 128.67, 128.14, 127.47, 122.05, 115.06, 114.98, 70.10, 68.31, 31.94, 29.93, 29.84, 29.80, 29.75, 29.71, 29.68, 29.62, 29.60, 29.43, 29.38, 29.34, 26.07, 22.71 and 14.14 ppm. C_36_H_49_NO_2_ requires: C, 81.90; H, 9.37; N, 2.65% found: C, 81.63; H, 9.48; N, 2.91%. MS M^+^ at *m*/*z* 527.68 (51%).

## Data Availability

Not applicable.

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
