# Peer review of "Synthesis of New Liquid-Crystalline Compounds Based on Terminal Benzyloxy Group: Characterization, DFT and Mesomorphic Properties"

_molecules, 2023, doi:10.3390/molecules28093804_

Round 1

Reviewer 1 Report

The paper of Mohamed A. El-Atawy and co-authors is an interesting fundamental work on synthesis three new Schiff bases, determining their structure using IR, NMR spectroscopy and mass spectrometry, studying phase transitions using DSC. Quantum chemical calculations were also applied to determine HOMO and LUMO and some other parameters. The introduction is well written, from which it can be understood that this work is a continuation of previously published results. The compounds obtained by the authors are new, which confirms the search in chemical databases. However, I have a few clarifying questions.

Most of the manuscript is quantum-chemical calculations in the gas phase. And from the text of the manuscript it is not clear how quantum chemical calculations explain the behavior of substances in the DSC experiment (Line 95: All possible structures are optimized using DFT to correlate with experimental results.)

There are HOMO and LUMO in the calculations, but the calculation itself does not give anything new. For example, in additional materials to the article there are UV spectra, about which there is no information in the text of the manuscript. Usually, quantum chemical calculations are used to explain UV spectra. What can be said about the UV spectra using the presented calculations?

It is also not clear why the calculations were carried out in points 2.4.3. Chemical Reactivity Descriptors and 2.4.4. Molecular Electrostatic Potential (MEP), because the use of liquid crystals is not associated with their reactivity, but for the most part with physicochemical parameters during phase transitions.

What is the "diluting effect" on line 164? Possible decrease in ordering? 

In the experimental part, it is indicated that all products are crystals. Why not do a single-crystal X-ray diffraction analysis, then quantum-chemical calculations with structure confirmation would be relevant.

What is in table 1 ∆S/R?

In supplementary materials in the article there is also a mass spectra, need to add a few lines to the text of the manuscript

Minor remarks: pay attention to the "degree" sign. Superscript letters “o” are used in various places, the terminus (line 64), stiff core (line 66), nth (line 69), moirty (line 176), aor (Line 180), 

While searching the database reaxys came across this article, it may be useful for the authors to add it to the experimental part. https://link.springer.com/article/10.1007/BF00529366

(I am not the author and in no way affiliated with the authors of this article)

In general, the manuscript contains new information about the synthesis of three new substances, which is enough to accept an article in a journal. I recommend accepting the paper for publication in the Molecules after the manuscript has been amended.

Author Response

Dear Reviewer,

I would like first to thank the Referee for his valuable and accurate comments that helped us to revise our manuscript more thoroughly.  All his suggestions have been considered in the revised manuscript via "Track Changes" function in the Microsoft Word.

Most of the manuscript is quantum-chemical calculations in the gas phase. And from the text of the manuscript, it is not clear how quantum chemical calculations explain the behavior of substances in the DSC experiment (Line 95: All possible structures are optimized using DFT to correlate with experimental results.)

There are HOMO and LUMO in the calculations, but the calculation itself does not give anything new. For example, in additional materials to the article there are UV spectra, about which there is no information in the text of the manuscript. Usually, quantum chemical calculations are used to explain UV spectra. What can be said about the UV spectra using the presented calculations?

It is also not clear why the calculations were carried out in points 2.4.3. Chemical Reactivity Descriptors and 2.4.4. Molecular Electrostatic Potential (MEP), because the use of liquid crystals is not associated with their reactivity, but for the most part with physicochemical parameters during phase transitions.

Reply: Thanks for the reviewer for his valuable comments. Accordingly, all the discussion regarding DFT calculations has been rewritten. To relate the calculations with the liquid crystal properties.

What is the "diluting effect" on line 164? Possible decrease in ordering? 

Reply: Meaning less ordering of the terminal flexible chain length.

In the experimental part, it is indicated that all products are crystals. Why not do a single-crystal X-ray diffraction analysis, then quantum-chemical calculations with structure confirmation would be relevant.

Reply: In fact, solid X-ray is not important due to we study the liquid crystal phase not the crystal phase.

What is in table 1 ∆S/R?

Reply: Normalized entropy = entropy change divided to gas constant to become dimensionless and widely used in liquid crystal investigations.

In supplementary materials in the article there is also a mass spectra, need to add a few lines to the text of the manuscript

Reply: it has been addressed. 

Minor remarks: pay attention to the "degree" sign. Superscript letters “o” are used in various places, the terminus (line 64), stiff core (line 66), nth (line 69), moirty (line 176), aor (Line 180),

Reply: Done. 

While searching the database reaxys came across this article, it may be useful for the authors to add it to the experimental part. https://link.springer.com/article/10.1007/BF00529366

Reply: it has been addressed. 

Reviewer 2 Report

This article examines the effect of the terminal benzyloxy group on the mesomorphic properties of rod-like Schiff base liquid crystals. The authors synthesized a new series of these liquid crystals and used DSC and POM to investigate their molecular self-assembly. They found that the length of the alkyl side chain influences various thermodynamic parameters of the derivatives, and molecules I16 and I6 have distinctive electronic structures. The study provides a comprehensive analysis of the impact of the terminal benzyloxy group on mesomorphic properties of liquid crystalline materials, which could be helpful in developing new materials for various applications. However, more details on experimental procedures and characterization techniques are required for clarity. Overall, the study is appropriate for the journal's scope, but major revisions are necessary for publication.

1.     The author should include more Schiff base compounds in the introduction and highlight their role in mesogenic properties. The reviewer also provides four highly relevant references related to Schiff base compounds with mesogenic properties that could be added to the manuscript. New Journal of Chemistry 41 (11), 4680-4688; Journal of Fluorine Chemistry 147 (2013) 36–39; Journal of Materials Chemistry C 6 (7), 1844-1852; New Journal of Chemistry 41 (18), 9908-9917.

2.     The author should include 1HNMR, 13C NMR and mass spectra for the newly synthesized compounds in the supporting information section. This would provide more detailed information on the chemical structure of the compounds and would help to confirm their identity. Additionally, it would aid other researchers who may wish to reproduce the work in the future.

3.     The author should provide POM and DSC data for compounds I8 and I16 to enable a better comparison. This would provide a more comprehensive understanding of the mesomorphic behavior of the compounds and would aid in the interpretation of the results. It would also enable a more accurate comparison between the three compounds and would add to the overall scientific value of the manuscript.

4.     The author should provide Small-angle X-ray scattering (SAXS) and Wide-angle X-ray scattering (WAXS) data for all newly reported compounds. The inclusion of these data would provide further evidence for the type of mesogenic phase exhibited by the compounds and would help to support the conclusions drawn in the study.

5.     The authors provide a reference to support their choice of using the DFT approach with the B3LYP 6-311G(d,p) basis set for their computational calculations. This would provide the readers with a better understanding of why this method was chosen and would add to the overall scientific value of the manuscript. Additionally, including a reference would also demonstrate that the authors have thoroughly researched and considered the best computational methods for their study.

The quality of English in the manuscript is acceptable, but a few areas could benefit from some polishing. The language is generally clear and easy to understand, but there are instances of awkward phrasing, incorrect word usage, and grammatical errors that detract from the clarity and precision of the writing. With some additional editing and proofreading, the manuscript could be improved in terms of its readability and overall quality.

Author Response

Dear Reviewer,

I would like first to thank the Referee for his valuable and accurate comments that helped us to revise our manuscript more thoroughly.  All his suggestions have been considered in the revised manuscript via "Track Changes" function in the Microsoft Word.

  1. The author should include more Schiff base compounds in the introduction and highlight their role in mesogenic properties. The reviewer also provides four highly relevant references related to Schiff base compounds with mesogenic properties that could be added to the manuscript. New Journal of Chemistry 41 (11), 4680-4688; Journal of Fluorine Chemistry 147 (2013) 36–39; Journal of Materials Chemistry C 6 (7), 1844-1852; New Journal of Chemistry 41 (18), 9908-9917.

Reply: it has been addressed. 

  1. The author should include 1HNMR, 13C NMR and mass spectra for the newly synthesized compounds in the supporting information section. This would provide more detailed information on the chemical structure of the compounds and would help to confirm their identity. Additionally, it would aid other researchers who may wish to reproduce the work in the future.

Reply: it has been addressed. 

  1. The author should provide POM and DSC data for compounds I8and I16 to enable a better comparison. This would provide a more comprehensive understanding of the mesomorphic behavior of the compounds and would aid in the interpretation of the results. It would also enable a more accurate comparison between the three compounds and would add to the overall scientific value of the manuscript.

Reply: it has been addressed. 

  1. The author should provide Small-angle X-ray scattering (SAXS) and Wide-angle X-ray scattering (WAXS) data for all newly reported compounds. The inclusion of these data would provide further evidence for the type of mesogenic phase exhibited by the compounds and would help to support the conclusions drawn in the study.

Reply: In fact, SAXS and WAXS instruments are unavailable. However, the POM analyses confirm the type of the mesophases.

  1. The authors provide a reference to support their choice of using the DFT approach with the B3LYP 6-311G(d,p) basis set for their computational calculations. This would provide the readers with a better understanding of why this method was chosen and would add to the overall scientific value of the manuscript. Additionally, including a reference would also demonstrate that the authors have thoroughly researched and considered the best computational methods for their study.

Reply: it has been addressed. 

Comments on the Quality of English Language

The quality of English in the manuscript is acceptable, but a few areas could benefit from some polishing. The language is generally clear and easy to understand, but there are instances of awkward phrasing, incorrect word usage, and grammatical errors that detract from the clarity and precision of the writing. With some additional editing and proofreading, the manuscript could be improved in terms of its readability and overall quality.

The manuscript has been revised thoroughly to correct the English mistakes

Round 2

Reviewer 1 Report

Dear Authors, Editor,

Most of my recommendations were taken into account during the revision of the manuscript. The authors politely and fully responded to comments. I recommend accepting this work in current form.

I wish you good luck, scientific discussion and citations.

Best wishes

Reviewer 2 Report

The authors have effectively addressed the concerns I raised as a reviewer, and consequently, I recommend accepting the paper in its present form.